# Validity of Linear and Nonlinear Measures of Gait Variability to Characterize Aging Gait with a Single Lower Back Accelerometer

**DOI:** 10.3390/s24237427

**Published:** 2024-11-21

**Authors:** Sophia Piergiovanni, Philippe Terrier

**Affiliations:** Haute-Ecole Arc Santé, HES-SO University of Applied Sciences and Arts Western Switzerland, 2000 Neuchâtel, Switzerland; sophia.piergiovanni@gmail.com

**Keywords:** gait analysis, accelerometers, gait variability, nonlinear dynamics, aging, fall prevention, metronome walking

## Abstract

The attractor complexity index (ACI) is a recently developed gait analysis tool based on nonlinear dynamics. This study assesses ACI’s sensitivity to attentional demands in gait control and its potential for characterizing age-related changes in gait patterns. Furthermore, we compare ACI with classical gait metrics to determine its efficacy relative to established methods. A 4 × 200 m indoor walking test with a triaxial accelerometer attached to the lower back was used to compare gait patterns of younger (N = 42) and older adults (N = 60) during normal and metronome walking. The other linear and non-linear gait metrics were movement intensity, gait regularity, local dynamic stability (maximal Lyapunov exponents), and scaling exponent (detrended fluctuation analysis). In contrast to other gait metrics, ACI demonstrated a specific sensitivity to metronome walking, with both young and old participants exhibiting altered stride interval correlations. Furthermore, there was a significant difference between the young and old groups (standardized effect size: −0.77). Additionally, older participants exhibited slower walking speeds, a reduced movement intensity, and a lower gait regularity. The ACI is likely a sensitive marker for attentional load and can effectively discriminate age-related changes in gait patterns. Its ease of measurement makes it a promising tool for gait analysis in unsupervised (free-living) conditions.

## 1. Introduction

As individuals age, their ability to walk is significantly impacted by a convergence of biomechanical and physiological changes. These changes are a result of age-related physiological and neurological alterations, including decreased muscle strength, lower cardiorespiratory fitness, degeneration of the sensory system, impaired neuromuscular coordination, or diminished joint mobility [1]. Compared to young adults, older people exhibit distinct gait characteristics: notably slower preferred walking speeds, shorter stride lengths, longer double-support stance times, reduced push-off forces, and a more flat-footed landing [2,3]. 

These adaptations suggest a shift toward a more secure gait pattern. This cautious manner of walking requires increased attention and cognitive resources for motor control. Indeed, walking is governed by a balance between automatic processes—fast, efficient neural mechanisms requiring minimal conscious attention—and executive control processes, which are slower and demand significant cognitive resources [4]. Older adults increasingly rely on executive control processes to manage walking [4]. Neuroimaging studies have shown increased activation in the prefrontal cortex of older adults during walking tasks, supporting the notion of reduced automaticity [5,6].

Gait automaticity can be evaluated using dual-tasking tests, involving walking while performing another concurrent task. The aim is to evaluate how divided attention modifies gait performance [7]. Known as motor cognitive interference, these modifications are more pronounced in older adults, particularly in those with cognitive impairments, compared to their younger or cognitively healthy counterparts [8]. Competing demands for limited attentional or mental resources is a key factor in gait performance [9]. When engaged in dual tasking, older individuals may struggle to allocate additional attention to maintain stable gait patterns. Age-related decline in executive function, particularly in dimensions like attention and cognitive flexibility, contributes to increased difficulty in dual-task performance among older adults [10,11]. These dual-task-related gait changes have been shown to be predictive of future falls in older adults, highlighting the importance of assessing gait automaticity in fall risk evaluation and prevention strategies [12].

Another key characteristic of gait in older individuals is increased variability [3,13]. Gait variability refers to the natural variation in the spatial and temporal characteristics of walking [14]. Classically, linear statistical measures of gait variability, such as standard deviation or range, focus on quantifying the amount of variation in a set of gait parameters. In contrast, nonlinear approaches—as delineated in nonlinear dynamical system theory [15]—can reveal the structured temporal patterns underlying this variability, emphasizing its deterministic rather than random nature. Nonlinear tools, such as entropic measures, fractal analysis, or those developed for the study of deterministic chaos, are used to evaluate this temporal structure of variability [16,17]. Both linear and nonlinear strategies have been used effectively to describe gait variability in the older population and predict the likelihood of falls [13,18,19,20,21,22].

Because age-related decline in walking ability and gait performance increases the risk of stumbling and subsequent falls [23], it is crucial to identify gait changes in older adults. Analyzing age-related gait modifications not only helps with the early detection of fall risks but also aids in developing and evaluating intervention strategies to maintain safe mobility [24]. In addition, gait analysis also offers a means to monitor individual progress during rehabilitation, providing insight into the patient’s recovery trajectory [25,26,27]. Traditionally conducted using motion capture systems in laboratory settings, gait analysis has advanced with the advent of inertial sensors like accelerometers, enabling real-world gait quality measurements [28]. These developments underscore the potential of portable, accessible tools in public health, especially for older adults, to provide continuous, objective data that may signal subtle changes in gait or balance predictive of fall risk [29]. 

The assessment of gait parameters in ecological contexts presents unique challenges. The process requires the use of simple, non-intrusive devices designed for prolonged use over several days. As a result, a single point measurement cannot provide a comprehensive understanding of gait kinematics and kinetics. A recent systematic review highlights that traditional linear variability metrics, including step time and step length variability, are inadequately captured using a single accelerometer positioned at the lower back [30]. The problem may arise because these methods need to accurately record each gait event to evaluate variance without noise, something that is difficult with a single accelerometer. In addition, this systematic review emphasizes that while nonlinear measures can serve as valuable complements to linear measures of variability, there remains a need for more high-quality studies to confirm their validity.

Over the last decade, several high-quality studies have demonstrated that accelerometric measures are effective at predicting fall risk among older adults in their everyday home environments. By collecting acceleration-derived data over extended periods, researchers have evaluated gait quality and quantity, identifying measures that sensitively distinguish between individuals with high and low fall risks. Significant predictors have included a comprehensive range of gait parameters—such as walking speed, stride length and frequency, intensity, regularity (autocorrelation function (ACF) method), smoothness, symmetry, and complexity—derived from both linear and nonlinear analyses [19,31,32,33]. However, there remains a need to develop advanced methods for analyzing acceleration signals that capture the full spectrum of gait characteristics, particularly those accounting for gait automaticity, to enhance our understanding of balance deterioration.

The ACIER (attractor complexity index empirical rationalization) study endeavors to fill this gap by validating an approach based on the nonlinear analysis of acceleration signals measured at the lower back level. We sought to validate a novel method derived from the nonlinear dynamic analysis of chaotic systems (Lyapunov exponents). Initially intended to evaluate gait resilience to perturbations, the maximal Lyapunov exponents method has also demonstrated a robust ability to quantify the long-range correlations observed over consecutive strides [34]. This particular structure of variability, characterized by a scale-free organization of temporal variance, is also known as statistical persistence, 1/f noise, power–law correlation, or fractal pattern [35]. Recognizable in various physiological signals, fractal dynamics has been proposed as a signature of the elaborate complexity found in living organisms [36]. A recent study has shown that the variability of lower limb muscle activity, as recorded by surface electromyography, has a similar long-range correlation pattern to the time series of gait intervals. This finding sheds light on the neurological basis of gait complexity [37]. 

Although early research suggested that a decrease in gait complexity indicated neural aging or degeneration [36], more recent studies have nuanced this assumption. They suggest that gait complexity is not static and can be modulated by a range of experimental factors and disease conditions. For example, dual tasking changes gait correlation patterns in older adults, linking gait complexity to motor-cognitive interference [38]. In patients with Parkinson’s disease, treadmill walking increases gait complexity, unlike in healthy controls. This suggests that externally paced walking may demand fewer attentional resources than self-paced walking in these individuals [39]. When exposed to visual field perturbations, older adults reduce their gait complexity—a phenomenon absent in younger individuals—which suggests they allocate more attention to gait control when confronted with external challenges [40]. People with chronic low back pain also exhibit reduced gait complexity when concentrating on their symptoms but see improvements when their attention is diverted [41]. Lastly, healthy young adults experience reduced gait complexity when performing cognitive tasks on smartphones while walking [42]. These findings collectively support the hypothesis that measuring the correlation patterns between successive strides can provide valuable insights into how much attention is allocated to gait control.

In the ACIER study, the use of metronome walking was proposed as a paradigm to induce increased attentional demands—or decreased gait automaticity—during walking. The process of aligning rhythmic auditory cues with motor actions (sensorimotor synchronization) relies on conscious intent; synchronization requires that a person must intend to move, indicating the mobilization of executive functions [43]. Sensorimotor synchronization is known to involve extensive cortical, subcortical, and cerebellar networks [44,45]. Metronome walking, where individuals match their steps to a steady beat, is known to alter the correlation pattern in stride intervals, resulting in an anti-correlated (or anti-persistent) pattern [46,47]. Studies have observed a relationship between the amount of attention paid to gait control and the degree of stride anti-correlation in treadmill experiments [48,49]. It appears that executive function is involved in the regulation of anti-persistence in the variable relevant to treadmill walking goal attainment [49]. Notably, this shift towards anti-persistence is also evident in other controlled walking tasks, such as synchronizing steps to floor markings or walking on a treadmill at a fixed speed [47,50,51]. These findings indicate that the observed decrease in gait complexity, shifting from correlation to anti-correlation, is more widely related to the attention allocated to gait control, rather than specifically tied to auditory-motor coupling.

In the initial phase of the ACIER study, our goal was to evaluate, in older people, the responsiveness of the proposed method (attractor complexity index, ACI) to metronome walking relative to an established reference technique (detrended fluctuation analysis, DFA) [52]. This involved optimizing the parameterization of the ACI algorithm and examining its intrasession reliability. The present article describes the second phase of the study, in which our focus shifted to determining the efficacy of this new method in characterizing the gait patterns of older individuals through comparative analyses with a younger cohort. In addition, we investigated the effectiveness of ACI relative to other established gait variability metrics. A central hypothesis tested was the unique sensitivity of ACI to metronome walking, which would induce distinct changes not observable with traditional variability parameters. The secondary hypothesis was that ACI could discriminate age-related changes in gait patterns as efficiently as other gait metrics.

## 2. Materials and Methods

### 2.1. Study Rationale and Design

In the context of identifying older individuals at increased risk of falling in their daily environment, the rationale behind the ACIER study centered on the use of a single triaxial accelerometer attached to the lower back, a methodology that represents the norm among researchers in this field [29,31,33]. To specifically validate our novel approach (ACI [34,52,53]), metronome walking was used to alter the correlation structure of gait and thus confirm the specific sensitivity of the ACI to attentional demands in gait control.

While our long-term purpose is to use ACI in unsupervised settings, we first needed to analyze its properties in more controlled contexts. Therefore, our validation study used a standardized indoor walking circuit to reduce environmental biases. In addition, the indoor setting facilitated accurate measurement of average walking speed, allowing for a more nuanced interpretation of the collected gait metrics and their interrelationships. 

The second phase of the ACIER study used a cross-sectional design and aimed to compare younger adults with the older adults enrolled in Phase 1 [52]. As in Phase 1, in addition to the lumbar accelerometer, the younger participants wore a second accelerometer attached to the foot, which served as a reference standard for calculating the correlation structure between strides. In addition to calculating both short- and long-term logarithmic divergence exponents (using the Lyapunov exponent method), several other gait metrics were derived from the lumbar acceleration signals for comparative analysis. Linear mixed-effects models were used as the method of inference to examine the effects of age group (older versus younger), walking condition (metronome versus normal), and walking speed as well as potential interactions between these factors. 

For additional insights into the ACIER study, particularly the study rationale, settings, recruitment and characteristics of older participants, detailed experimental procedures, and an in-depth explanation of the ACI algorithm, readers are encouraged to refer to the companion article [52].

### 2.2. Participant Recruitment and Eligibility 

In addition to older adults already enrolled in Phase 1, our goal was to recruit 40 healthy adult participants between the ages of 18 and 40 without gait disorders of orthopedic or neurologic origin. The minimum age limit was set in response to legal restrictions, thus excluding minors. The maximum age limit was determined based on epidemiologic [54,55] and experimental [56,57] studies that have shown that changes in both static and dynamic balance can occur in individuals in their forties or fifties, similar to those observed in older adults. Young participants were recruited through a combination of personal networks and electronic advertisements. The research team used connections within our institution (HE-Arc), including students, colleagues, friends, and family, supplemented by targeted email campaigns to partner institutions and internal mailings within HE-Arc. This approach aimed to ensure a diverse pool of participants. 

### 2.3. Experimental Procedures

Participants were equipped with two triaxial accelerometers (Physilog 6S, Gaitup, Lausanne, Switzerland): one attached to the lower back at the L4–L5 level and the other attached to the instep of the right foot. They walked a 205-m corridor four times at their natural pace, completing two round trips (Figure 1). Each segment was timed to measure their preferred walking speed. After the first round trip, there was a five-minute break during which walking cadence (step frequency, SF) was calculated from the lower back acceleration signal. On the next round trip, participants adjusted their gait to an electronic metronome, the rhythm of which was matched to the cadence determined during the break. A 30-s walk with the metronome was performed before the start of the second round trip to familiarize participants with gait synchronization. Like the older participants, the young participants wore their own low-rise, comfortable shoes during the walking test; high heels were not allowed.

### 2.4. Data Analysis

Acceleration data—sampled at a rate of 256 Hz—were transferred from the devices to the institutional server. Calculations and analyses were performed in MATLAB R2021a (The MathWorks Inc., Natick, MA, USA). No prefiltering of the acceleration signals was performed prior to analysis. Segments indicative of steady walking were selected for subsequent analysis. Specifically, non-walking periods were discarded, including the time between the activation of the accelerometers and the onset of walking, the 5-s break participants took midway during their walk along the 200-m corridor, and the time between the end of the walk and the deactivation of the accelerometers. These non-walking periods were identified and excluded by manually reviewing the acceleration data using a MATLAB-generated figure. Raw acceleration data are available online [58]. 

Table 1 presents a summary of the gait metrics derived from the acceleration signals (with the exception of the preferred walking speed, which was assessed by timing the walks). The first column briefly outlines the methodology, explains its basic rationale, and cites key papers that provide comprehensive descriptions of the algorithms used. The subsequent column references studies that have applied the metric in unsupervised settings. 

To strengthen the validity of gait metrics for unsupervised settings where direct speed measurement is not possible, we measured average preferred walking speed by timing participant displacement. An analytical framework was then developed to categorize gait metrics based on their association with speed and responsiveness to age. Some gait metrics may characterize aspects of gait decline that are unrelated to the expected decline in preferred walking speed. No correlation with walking speed was expected. These have been termed “speed-independent” metrics. Conversely, “speed-surrogate” metrics show a strong correlation with walking speed, with this association expected in both younger and older adults. Finally, “mixed metrics” represent an intermediate category. They show a correlation with speed, but this correlation is likely to be stronger in older adults than in younger individuals. Gait aspects characterized by mixed metrics are hypothesized to be predominantly associated with the age-related decline in preferred walking speed.

We defined “basic gait parameters” as commonly used metrics that describe the gait cycle during ambulation, including step length, step frequency (SF), and walking speed. We excluded step length from the analysis to avoid redundancy, as it can be derived from speed and SF. Average SF was calculated using spectral analysis (fast Fourier transform) of the vertical acceleration from the lumbar accelerometer, identifying the dominant frequency within the signal. We determined the average preferred walking speed by timing participants as they walked the 205-m corridor. After calculating the average SF, we calibrated the raw 3D acceleration signals using the method proposed by Moe-Nilssen [69]. This algorithm corrects accelerometer data for tilt to obtain meaningful acceleration information in a horizontal–vertical coordinate system. In addition, we used an alternative method to generate gait metrics that are robust to orientation or displacement problems with the motion sensor. The vector norm (or magnitude) of the 3D acceleration signals was calculated as the square root of the sum of the squares of the individual acceleration components. Note that to be consistent with the Moe-Nilssen procedure, the effect of Earth acceleration was removed by subtracting one of the raw vector norms.

To standardize gait metrics across participants, we truncated acceleration signals to 250 steps (125 strides). This was accomplished by multiplying the individual step frequency (SF) by the accelerometer sampling rate (256 Hz) and the desired step count (250). The resulting signals in anteroposterior (AP), vertical (V), and mediolateral (ML) directions and the vector norm varied in duration but represented an equal number of steps. These standardization procedures were not applied to foot accelerometer data (see below).

To assess the intensity of the walking movements, the RMS of the vector norm was determined over the 250 steps. It is well known that gait RMS has a significant correlation with walking speed [70] and thus can serve as an effective proxy measure for this parameter in unsupervised conditions (speed-surrogate metric). It has been suggested that standardization of the root mean square (RMS) may provide a linear variability index that is less sensitive to gait speed but more indicative of gait abnormalities [62]. Consequently, the RMS ratio is defined as the quotient of the RMS of the acceleration signal measured in the mediolateral direction and the RMS of the vector norm. 

To quantify gait pattern regularity, we performed an unbiased autocorrelation analysis using the vector norm as the input (ACF method [64]). Autocorrelation measures the similarity between a signal and its delayed version over various time intervals. We utilized Matlab’s xcorr function with the “unbiased” option. The autocorrelation sequence was normalized by dividing each element by the zero lag value, scaling results between −1 and 1, akin to a correlation coefficient. The first and second dominant periods correspond to single step and stride phase shifts, respectively. Maximal values at these peaks define step and stride regularity. To enhance statistical robustness, we applied Fisher’s transform to normalize values following Auvinet et al.’s recommendations [71].

We calculated logarithmic divergence exponents using the maximal Lyapunov exponent method and Rosenstein’s algorithm as nonlinear measures of gait stability and automaticity [65,66,67]. The companion article provides detailed computation methods for short-term (LDS) and long-term (ACI) divergence exponents. Briefly, we standardized acceleration signals to 18,750 samples (75 per step), constructed a multidimensional attractor using Takens’ theorem, calculated logarithmic divergence between adjacent trajectories, and determined divergence rates over 0–0.5 strides (LDS) and 5–12 strides (ACI). Following previous recommendations [52,72], we analyzed LDS along the mediolateral axis (LDS-ML) and ACI along the anteroposterior (ACI-AP) and vertical (ACI-V) axes as well as for the vector norm (ACI-N).

A secondary accelerometer was attached to the foot to improve the accuracy of gait event identification, allowing for a more accurate stride interval time series. This improvement was critical for DFA, the reference method for assessing gait correlation structure, which requires a discrete time series of gait events. The companion article provides a comprehensive rationale and detailed explanation of this procedure [52]. Briefly, a peak detection algorithm applied to the foot acceleration signal delimited individual strides and calculated their durations. The resulting stride interval time series was analyzed using DFA, with box sizes ranging from 16 to N/2 (where N is the total number of strides). The DFA analysis yields a scaling exponent (α) that indicates statistical persistence (fractal pattern) and high complexity if between 0.7 and 1 or an anticorrelated (anti-persistent) pattern if below 0.5.

### 2.5. Statistics

Our approach to characterizing gait metrics used a combination of visualization, descriptive statistics, and further exploration techniques. For better interpretation, the outcomes of the two segments (back and forth) were aggregated (mean). Visualization used boxplots and univariate scatterplots (Figure 2, Figure 3 and Figure 4) to initially explore the distribution of the data. Descriptive statistics summarized central tendencies and variability for both normal and metronome walking conditions across age groups, presented as means and standard deviations in Table 2 and Table 3. These tables also include standardized differences between young and older participants (Hedges’g [73]) along with 99% confidence intervals (CIs) calculated using bootstrapping. Further exploration, detailed in the Appendix A, included histograms and normal probability plots as well as bivariate scatterplots with linear fits and corresponding Pearson’s correlation coefficients, allowing for more in-depth examination of data relationships.

We employed linear mixed-effects models to analyze the impact of age and walking conditions on gait metrics [74]. This method accounts for correlations in repeated measurements for each participant and the nested data structure, as each condition involved two walking segments (Figure 1). After visually inspecting histograms and probability plots to confirm normality and removing outliers above three standard deviations, gait metrics were set as dependent variables. Age group (older vs. young) and walking condition (metronome vs. normal) were included as fixed effects through dummy variables. The model’s random component included a slope effect to capture individual variability in response to walking condition (participant-by-condition interaction). To evaluate if age groups responded differently to the metronome, we included a group-by-condition interaction term as a fixed effect. Using maximum likelihood estimation, F-tests assessed the significance of fixed effects (ANOVA method). If the interaction term was not significant (*p* > 0.01), it was excluded, and the model was refitted with restricted maximum likelihood. Detailed analysis results are in the Appendix A, and Table 4 presents fixed-effects coefficients with 99% confidence intervals. 

To compare effect sizes across gait measures, we performed additional analyses using mixed-effects models with standardized coefficients. We standardized the dependent variables by subtracting the mean and dividing by the standard deviation, allowing fixed-effects coefficients to be expressed in standard deviation units for direct comparability. As in the initial analyses, the models included dummy variables for group and condition along with random slopes for the condition-by-participant interaction. To refine our results, we included preferred walking speed as a continuous covariate, helping to clarify distinctions among the three gait metric categories (speed-independent, speed-surrogate, and mixed). Results are shown in Figure 5.

In our analysis, 10 gait metrics derived from lumbar and foot accelerometers were analyzed using linear mixed-effects models. These gait metrics are expected to exhibit some degree of interdependence. To account for the potential inflation of Type I error due to the increased number of parallel statistical tests, we chose a significance threshold of 0.01. We believe that a more stringent correction method such as Bonferroni, which assumes independence between tests, may be overly conservative in this context. A 0.01 threshold balances the need to control for false discovery while maintaining sufficient statistical power (reducing Type II error).

The second phase of the ACIER study aimed to recruit 100 participants (60 older adults, 40 young adults). As detailed in the companion article on the first phase [52], prior treadmill experiments suggested a strong metronome effect on ACI, with an effect size > 2 [67], indicating that a small sample (N < 10) would suffice to detect this primary effect. Our secondary goal was to examine age-related differences in ACI, guiding our sample size calculation. Due to limited data on ACI variability and age responsiveness, we referenced studies using the scaling exponent from DFA. However, a meta-analysis of eight DFA studies [20] showed significant heterogeneity in scaling exponents and age effects with effect sizes from −1.26 to +1.14 (mean ES = −0.20), likely due to differing measurement methods and algorithm parameters. Moreover, most included studies focused on older adults younger than our sample of older participants. The study with the oldest participants (mean age 76) reported an effect size of −1.26 [75]. Another recent study, with a robust methodology, compared 23 older adults (mean age 72) to 22 younger controls and reported an effect size of −0.91 [76]. Given these findings, we conservatively set an expected effect size of −0.7. To achieve 80% power with a 1% alpha level and a minimum effect size of −0.7, we calculated a sample size of 90 (36 young, 54 older). We increased this to 100 participants (60 older, 40 young) to account for potential data loss and dropouts.

## 3. Results

### 3.1. Participants

Between the second semester of 2022 and June 2023, 42 young adults, including 16 males (38%) and 26 females (62%), participated in the indoor walking test. Their mean age was 27 years (SD = 5.9), with a body mass of 68.9 kg (SD = 17.7) and height of 1.71 m (SD = 0.08). For the older adult group of Phase 1 (N = 60), the mean age was 76 years (SD = 6), body mass was 74 kg (SD = 16), and height was 1.68 m (SD = 0.08), with 60% females (*n* = 36) and 40% males (*n* = 24). 

### 3.2. Data Visualization and Cleaning

There were no missing data for gait parameters within the young adult group. Detailed breakdowns of sample size for each variable are presented in Table 2 and Table 3.

Overall, the dependent variables showed no large deviations from normality, as revealed by boxplots, scatter plots, and histograms (Figure 2, Figure 3 and Figure 4 and Appendix A). However, speed, RMS, and RMS ratio exhibited some degree of skewness when probability plots were considered (Appendix A). We consider these minor deviations acceptable because linear mixed models are robust against violations of normality assumptions [77]. No data points were identified as outliers.

### 3.3. Descriptive Statistics

#### 3.3.1. Age Effects

Examination of differences between the older and the young group (Table 2 and Table 3) revealed that older people walked under non-cued (normal) condition with a lower speed (ES = −0.80) and lower movement intensity (RMS ES = −0.58). They also had lower step regularity (ES = −0.97) and stride regularity (ES = −0.91) as well as a lower ACI along both axes (ES = −0.77 and −0.69) and when considering the vector norm (ES = −0.53). Standardized differences between age groups measured from the metronome walking condition were mostly slightly higher: speed ES = −0.77; step regularity ES = −1.02; stride regularity ES = −1.01; ACI-AP ES = −0.95; ACI-V ES = −0.92; ACI-N ES = −0.87. In contrast, movement intensity, as measured by RMS, showed a lower contrast between age groups in metronome walking (RMS ES = −0.46). Other gait metrics showed smaller differences between age groups, with absolute ES values not exceeding 0.38.

#### 3.3.2. Metronome Effects

Examination of the differences between the normal and metronome walking conditions showed that only the ACIs and scaling exponents (DFA) were substantially altered. For the older group, the relative changes were: ACI-AP = −27% (ES = −0.74), ACI-V = −26% (ES = −0.78), and ACI-N = −30% (ES = −0.82). The scaling exponent showed a relative difference of −47% (ES = −1.75). In the younger group, the relative changes were: ACI-AP = −14% (ES = −0.57), ACI-V = −15% (ES = −0.63), and ACI-N = −13% (ES = −0.52). The relative difference for the scaling exponent was −40% (ES = −1.79).

#### 3.3.3. Correlations

Analysis of correlations between gait metrics across groups and conditions revealed some recurrent patterns of associations (Appendix A). For young participants, we found moderate correlations between ACI and scaling exponents when both conditions were aggregated, aligning with previous findings in older adults [52]. The aggregated results (N = 84) were ACI-N r = 0.45; ACI-AP r = 0.45; ACI-V r = 0.45. Similarly, in the metronome walking condition (N = 42), moderate correlations were observed (ACI-N r = 0.51; ACI-AP r = 0.49; ACI-V r = 0.46). However, in the normal walking condition, the correlations were weaker and not significant (*p* > 0.05). The normal condition data (N = 42) showed ACI-N r = 0.21; ACI-AP r = 0.18; ACI-V r = 0.19. When analyzing the entire dataset (both groups and conditions combined), the correlations between ACI and scaling exponent were significant, even with a stricter threshold (*p* < 0.01). The combined data (N = 204) yielded ACI-N r = 0.52; ACI-AP r = 0.45; ACI-V r = 0.50.

Unlike older participants [52], young participants did not exhibit significant correlations between ACI and walking speed for either axis (r = −0.19 to 0.11) or the norm (r = 0.00 to 0.10) across conditions. Similarly, for older participants, a robust positive association was observed between ACI and stride regularity as well as step regularity (r = 0.58 to 0.68). In contrast, this association was weaker in younger participants (r = 0.16 to 0.31).

In examining other relevant associations between gait metrics and preferred walking speed, stride frequency (SF) showed a moderate to strong correlation with speed (r = 0.43 to 0.69 across groups and conditions), consistent with expectations. Similarly, movement intensity (RMS) was robustly correlated with speed (r = 0.73 to 0.79). In contrast, gait parameters such as scaling exponents (r = −0.10 to 0.19), LDS (r = −0.28 to 0.04), and ACI-AP (r = −0.19 to 0.34) exhibited negligible associations with walking speed. Furthermore, differences were observed between age groups. The older participants showed a moderate correlation between speed and step regularity (r = 0.42 to 0.58), while the younger participants showed minimal to no correlation (r = 0.00 to 0.21).

### 3.4. Inferential Statistics

The primary hypothesis of this study was that ACI would be specifically sensitive to metronome walking, in contrast to other gait parameters measured by the lumbar accelerometer. Linear mixed-effects regression analyses confirmed this hypothesis. When controlling for age group, models that included the anteroposterior axis (ACI-AP), vertical axis (ACI-V), and norm (ACI-N) exhibited significant regression coefficients (*t*-test, *p* < 0.0001; Table 4 and Appendix A). This conclusion held true even after adding walking speed as a covariate (Figure 5). Scaling exponents obtained from the foot accelerometer, used as a reference metric, showed similar responsiveness to metronome walking (Table 4 and Figure 5). Conversely, other gait metrics remained largely unaffected by synchronization to the metronome. However, movement intensity (RMS) exhibited significant sensitivity to the metronome (*p* = 0.002), although the association was very weak (Figure 5). The regression model predicted a relative difference of 3.8%.

The secondary objective of this study was to assess the ability of the ACI to characterize aging gait patterns. Regression models (Table 4 and Appendix A) demonstrated a significant effect of age groups on ACI-N (*p* < 0.001), ACI-AP (*p* < 0.0001), and ACI-V (*p* < 0.0001). Figure 5 illustrates the effectiveness of the ACI in discriminating between older and younger adults compared to other gait metrics. The figure displays standardized coefficients that indicate the strength of association, adjusted for speed and condition. These coefficients are ranked from highest (top) to lowest (bottom) group effect. Step and stride regularity and ACI-AP exhibited similar magnitudes. ACI-V and ACI-N showed slightly lower beta coefficients. The remaining gait metrics did not show a significant response to age groups. However, note that the wide confidence intervals (Table 4 and Figure 5) may suggest insufficient statistical power, which could potentially mask truly relevant effects.

Finally, to further explore the responsiveness of gait metrics, we investigated their associations with preferred walking speed. Figure 5’s right subplot presents the covariances between walking speed and each gait metric after adjusting for age and condition effects. Movement intensity (RMS) and SF were the parameters with the strongest correlations with speed. Weaker yet significant associations were observed for stride and step regularity as well as ACI-V and ACI-N. Conversely, ACI-AP, LDS, scaling exponent, and RMS ratio did not show significant associations with preferred walking speed.

## 4. Discussion

The second phase of the ACIER study was designed to test the hypotheses that (1) the novel ACI metric is sensitive to changes in attentional demands during walking tasks compared to other linear and nonlinear gait metrics and (2) the ACI metric differs between young and older adults. Our results support the primary hypothesis by demonstrating a specific sensitivity of the ACI compared to other gait metrics derived from the lumbar accelerometer when participants walked in synchrony with a metronome compared to normal walking. Consistent with the secondary hypothesis, the ACI, along with step and stride regularity, demonstrated a significant ability to discriminate between older and younger adults.

### 4.1. ACI and Metronome Walking

Building on the responsiveness of the ACI to metronome walking observed in older adults (Phase I [52]), this study extended these findings to young adults (Table 2, Table 3 and Table 4 and Figure 5). We also found a significant correlation between the ACI and the reference standard (scaling exponent of the stride interval time series) in young adults (see Appendix A). These findings are consistent with prior studies showing similar responsiveness to voluntary synchronization and correlations with the reference standard (DFA) in both treadmill [34,53,67,78] and overground [79] walking studies. Our previous modeling study demonstrated a 66% reduction in ACI when comparing correlated versus anticorrelated time series after artificially altering real gait acceleration signals [34]. Descriptive statistics (Table 2 and Table 3) revealed a trend for younger individuals to have smaller differences in ACIs between normal and metronome walking compared to older adults (−26% to −29% vs. −12% to −15%, Table 2 and Table 3). This age-related discrepancy is consistent with the observed differences in scaling exponents (Table 2 and Table 3, young: −40%, older: −47%). This finding may parallel observations in treadmill walking, where lower scaling exponents are associated with tasks requiring tighter gait synchronization and potentially higher brain processing load [48]. Similarly, in our study, synchronizing gait with the metronome may demand more attention from older adults than younger participants. However, the regression analyses considering within-subject correlations did not show a significant condition-by-group interaction (*p* > 0.05, Appendix A, ANOVA F-tests). This lack of interaction may be due to insufficient statistical power, as detecting interaction effects often requires much larger sample sizes [80]. Future studies with adequate power are needed to explore this potential age effect.

### 4.2. Other Gait Metrics and Metronome Walking

Our study’s key contribution is the specific sensitivity of ACI to metronome walking, which sets it apart from other gait variability measures typically used in unsupervised conditions. Specifically, walking in synchrony with an isochronous metronome tuned as the preferred SF did not impact gait regularity, as assessed by ACF. Likewise, other gait metrics, including the RMS ratio and LDS, were unaffected by metronome walking. However, we observed a small (+4%) but significant increase in movement intensity (RMS) during this condition, even after accounting for potential within-participant correlations and adjusting for age group and speed using mixed effects models (Figure 5). While a type I error cannot be completely excluded, a plausible explanation is that participants may have unconsciously adjusted their gait by striking their heels more forcefully to match the metronome, resulting in higher peak accelerations and a slight increase in RMS.

### 4.3. Preferred Walking Speed and Age Effects

Walking speed is a valuable health biomarker [81]. Age-related conditions can limit gait speed in older adults. These conditions include sarcopenia (muscle loss [82]), reduced cardiorespiratory fitness [83], increased energetic cost [84], mitochondrial dysfunction [85], joint stiffness [86], or impaired sensory feedbacks [87]. As a result, older adults typically have a slower preferred walking speed than younger adults, which was confirmed in our study. Indeed, we found that older participants had an average speed of 1.27 m/s (Table 2 and Table 3), which is consistent with the reference value of 1.30 m/s reported for highly functional older individuals (mean age 79 years) in a recent large-scale study with a comparable socio-cultural context (Germany, [88]). In contrast, the walking speed of the younger participants was 1.43 m/s, which is slightly higher than the reference value of 1.37 m/s for adults between 20 and 40 years of age calculated from the data of a meta-analysis including 23,000 individuals measured worldwide [89]. Overall, we observed an 11% relative difference between young and older participants. This difference was highly significant in both the descriptive statistics (standardized ES: −0.77) and in the multiple regression analysis adjusted for walking condition (regression coefficient b = −0.167 m/s, *t*-test *p* < 0.0001, Table 4 and Appendix A).

### 4.4. Movement Intensity and Age Effects

In our effort to establish an analytical framework for categorizing gait metrics based on their correlation with observed preferred walking speed, we anticipated that both movement intensity (RMS of lumbar acceleration) and SF would fall into the speed-surrogate category. Single lumbar accelerometer gait analysis often utilizes simple gait models like the inverted pendulum to approximate walking speed (Table 1). Inverted pendulum models are typically derived through a double integration procedure of the trunk acceleration to estimate step lengths [90]. It is noteworthy that RMS captures the average amplitude of the signal, which is somewhat analogous to integrating the signal. In experimental studies involving subjects walking across a wide spectrum of speeds, a curvilinear relationship was observed between acceleration RMS and walking speed [70,91]. Our results support this finding, showing a strong correlation between RMS and preferred walking speed (correlation coefficients: 0.73–0.79, Appendix A) across age groups. We confirm therefore that older participants exhibit lower RMS values compared to their younger counterparts (Table 2 and Table 3). The regression model shows a significant effect for age group adjusted for walking conditions (Table 4), with the age effect disappearing when adjusted for walking speed (Figure 5), consistent with the strong RMS–speed relationship. However, while RMS shows promise as a walking speed surrogate, its sensitivity to factors beyond velocity, like surface type [92] and slope [93] suggests caution in using it as a robust proxy in unsupervised settings.

### 4.5. Step Frequency and Age Effects

As a second speed-surrogate metric, SF, which is biomechanically related to walking speed as the product of cadence and step length, also showed a systematic correlation with walking speed (Appendix A). However, the strength of this association across groups and conditions (r = 0.43 to 0.69) was weaker than that observed for RMS (r = 0.73 to 0.79). In contrast to RMS, SF did not differ significantly between young and older participants (1.89 vs. 1.90), as confirmed by the multiple regression analyses (Table 4 and Figure 5). This finding is consistent with the expected pattern in which older adults typically have shorter step lengths compared to younger adults [2], resulting in a similar cadence despite their reduced preferred walking speed (Table 2). Although SF may serve as a surrogate for walking speed, its utility in characterizing age-related gait changes in unsupervised settings seems therefore limited. However, more research is needed to further evaluate the usefulness of SF in assessing fall risk.

### 4.6. RMS Ratio and Age Effects

To attenuate the speed sensitivity of RMS, the RMS ratio has been proposed as a metric for gait quality assessment [62]. This approach aimed to normalize the mediolateral acceleration RMS by the global RMS of the vector magnitude, potentially revealing gait instability in the frontal plane. While several studies have investigated this hypothesis, the results have been mostly inconclusive [63,94,95]. In our large study including 100 subjects of various ages, we found no significant change in RMS ratio across the lifespan [56]. Here, too, we observed that RMS ratio was not different between young and older participants (Table 2, Table 3 and Table 4 and Figure 5). Although the concept of RMS normalization initially appeared promising, the use of the RMS ratio to characterize age-related gait changes seems to have limited utility.

### 4.7. Gait Regularity and Age Effects

Our results show a significant difference in step and stride regularity between age groups, as measured by the ACF analysis. The ES for normal walking was large (−0.97 and −0.91, Table 2), which was further supported by the results of the multiple regression analyses (Table 4 and Figure 5). This is consistent with previous research highlighting similar age-related reductions in regularity [96,97]. In particular, the previous work by Auvinet et al. [98], which characterized gait patterns across different age groups, reported a substantial difference in stride regularity (standardized ES = −1.22) between the youngest (20–29 years) and oldest (>70 years) cohorts, although no significant change was found across the lifespan. This observed change in acceleration signal regularity with age suggests that a convergence of musculoskeletal and neurological deficits may limit the ability to maintain consistent gait patterns across strides. Note also that a correlation was observed between ACI and ACF measures in the older group specifically (Appendix A, r = 0.58 to 0.68). This supports the hypothesis that both gait regularity and ACI are concomitant signs of gait quality degradation in older adults. Regarding the categorization of step and stride regularity according to their associations with walking speed, they can be considered as mixed metrics. Indeed, while only weak correlations were observed in the young group (r = 0.0 to 0.36, Appendix A), stronger associations were observed in the older group (r = 0.42 to 0.58). 

### 4.8. Local Dynamic Stability and Age Effects

LDS is a popular nonlinear gait variability metric that reflects gait robustness to perturbations and is considered a predictor of fall risk [22,66]. Unlike the ACI, where higher values indicate better gait quality, higher LDS results correspond to greater instability and a degraded gait pattern [19,34]. Therefore, it is expected that older adults, who have a higher risk of falling, will exhibit higher short-term divergent exponents compared to younger individuals. This assumption is partially supported by observational studies. However, definitive conclusions are hindered by methodological inconsistencies in measurement methods, algorithm implementations, and experimental designs [21]. Our 2015 study investigated changes in gait stability between the ages of 20 and 70 in 100 participants using the same LDS calculation as this work. Our model predicted a 13% higher LDS-ML at age 75 than at age 25 [56]. Previous studies have also reported varying relative differences between younger and older cohorts: +40%, graphical estimation [99]; +10% [100]; +7% [101]; +6%, non-significant [102]; +5%, non-significant [103]; and +2%, non-significant [40]. The observed relative difference in the present study was +14% during normal walking and +8% during metronome walking (Table 2 and Table 3), which is consistent with previous research. However, the null hypothesis of no difference between age groups could not be rejected due to large confidence intervals (Table 4 and Figure 5). Further research is necessary to specifically determine the discriminatory power of the LDS in identifying individuals at risk of falls compared to those who are not.

### 4.9. Attractor Complexity Index and Age Effects

Our findings (Table 2, Table 3 and Table 4, Figure 5) strongly support the hypothesis that the ACI can distinguish age-related changes in the gait patterns. This new discovery has no direct precedent in the literature given that ACI has only been recently proposed as a gait metric for replacing DFA in gait analyses under free-living conditions [34,52,53]. Based on the mixed-effects models outcomes (Table 4), we can predict a relative change of −24% between the young and older populations. To put this in perspective, we found a contrast of −18% between healthy adults and adults suffering from chronic pain of lower limbs using ACI (still referred to as LDS-L in this 2017 study [60]) in unsupervised gait analysis. This comparison is instructive because chronic pain patients may use greater voluntary control of limb movements to adopt a less painful gait (antalgic gait [4,41,104]), possibly similar to the increased cognitive interference observed in older adults. 

In the initial phase of the ACIER study, which focused exclusively on older adults, our analysis showed that the ACI measured in the frontal plane (ACI-ML) was inadequate for assessing gait correlation structure [52], while both ACI-AP and ACI-V could be used interchangeably. We further hypothesized that ACI-N, as an orientation-free metric, could be beneficial in unsupervised gait analyses, where maintaining a consistent sensor positioning can be challenging. However, including the results from the young adult cohort yielded a more nuanced interpretation. ACI-AP showed greater sensitivity to age-related changes than ACI-V and ACI-N (Figure 5). Additionally, ACI-AP emerged as a speed-independent metric, whereas ACI-V and ACI-N correlated with walking speed, especially in older adults (Appendix A). Although the pronounced sensitivity of the ACI-AP warrants further investigation, it is plausible that acceleration pattern within the sagittal plane—aligned with the displacement of the body—may hold information for the regulation, and thus for the correlation structure, of stride length and duration. In contrast, such regulatory information may be less apparent in accelerations occurring perpendicular to the direction of movement.

### 4.10. Scaling Exponent and Age Effects

In contrast to the observed age-related differences in ACI, scaling exponents derived from foot acceleration data using DFA did not show significant age group differences in the mixed model analysis (*p* = 0.06, Appendix A, *t*-test). This lack of significance may be due to limitations of the DFA method, as highlighted in our companion article regarding its potential inaccuracy at short measurement times [52]. Although the observed standardized difference between age groups in normal walking (effect size = −0.19, Table 2) lacks statistical significance, it aligns with the findings of the above-mentioned meta-analysis (ES = −0.20, combining eight studies [20]). This suggests that ACI (ES = −0.77, Table 2) may be more effective than DFA in detecting age-related changes in gait patterns, particularly in shorter walking bouts or in unsupervised settings. However, further independent studies are needed to consolidate this observation.

### 4.11. Gait Metrics and Age-Related Decline in Walking Abilities

Overall, our findings reinforce the notion that reliance on a single metric may not adequately capture the complexity of age-related decline in walking abilities. A comprehensive suite of both linear and non-linear gait metrics derived from acceleration signals appears to be essential to identify the various age-related adaptations in gait patterns. We propose that the ACI-AP, considered as a speed-independent metric, could reveal subtle changes in the automated control of gait—a modification not necessarily associated with reduced walking speed. At the same time, metrics of step and stride regularity, which we categorize as mixed metrics, could reveal difficulties in maintaining a consistent gait pattern from one stride to the next. This inconsistency could be caused by a reduced lower limb strength, potentially leading in parallel to a reduced preferred walking speed.

## 5. Strengths and Limitations

The inclusion of community-dwelling older adults ages 65+ years is a key strength of our study, as this population represents both those at a high risk of falls and a prime target for primary fall prevention efforts, aligning with research showing the effectiveness of early intervention in active, independently living seniors [24,105]. The use of a standardized indoor circuit of substantial length (2 × 205 m) performed in a “real” building is also a strength of the study, as it provides a more ecologically valid setting than treadmill walking or short walks in a gait laboratory while still allowing for control over environmental factors that could introduce bias in outdoor settings. Additionally, this setup enabled the direct measurement of walking speed by timing participants’ walks. Finally, the use of metronome walking as a method to specifically increase the attentional load dedicated to gait control is also a positive point. Compared to traditional dual-tasking experiments used to manipulate attention [106], metronome walking can be sustained for longer durations without fatigue, allowing for more extended gait analyses. Additionally, metronome walking poses fewer safety risks for individuals prone to falling, as it does not introduce the same level of distraction or complexity as dual-task conditions [107].

Our study also had several limitations. The use of a metronome to increase attentional demand is not a standard practice for testing gait automaticity. Although the introduction provides strong evidence suggesting metronome walking may increase attentional requirement, this relationship needs further confirmation through additional research. The selection of gait metrics for comparison with ACI was based on their applicability in unsupervised contexts, as outlined in Table 1. While this choice was somewhat arbitrary and not exhaustive, it was necessary to limit the number of metrics to maintain a low false discovery rate given our sample size. Further research is warranted to comprehensively evaluate ACI’s performance against a broader range of gait metrics. The present study was limited by the relatively short duration of gait assessment and the absence of repeated measurements across different days and contexts (e.g., outdoor walking, fast walking, slope walking). While previous research has examined the reliability of ACI between measures separated by 9 days [108] and in unsupervised settings over 7 days of measurement [60], further validation in ecological contexts is warranted. Future studies should aim to assess ACI’s performance over longer durations and in a variety of real-world walking conditions to more comprehensively evaluate its utility as a gait analysis tool. 

## 6. Conclusions

In conclusion, ACI shows a unique sensitivity to metronome walking in both young and older adults. This suggests that ACI specifically captures changes in gait control associated with increased attentional demands during synchronized walking. Second, ACI appears to be a valuable tool for discriminating age-related differences in gait patterns. This finding is consistent with the observed differences in step and stride regularity, further highlighting the potential of ACI as a complementary marker of gait quality decline in older populations and thus as a tool for identifying older adults at risk for falls. In addition, due to the relative ease of measuring ACI, it could be used to evaluate practical interventions, such as in the recent clinical trial aimed at restoring gait automaticity in older adults that began in parallel with the ACIER study [109]. The broader implications of these findings go beyond fall risk assessment. The sensitivity of ACI to attentional load during gait opens new avenues for investigating the complex interplay between cognitive function and motor control of human locomotion, which may help to gain deeper insights into the mechanisms underlying gait disorders.

Building on these promising findings, the next phase of the ACIER study will focus on the clinical utility of the ACI for fall risk assessment. We will use a retrospective approach to differentiate between participants who have recently fallen and those who have not. This will involve the analysis of gait data collected in the first phase of the ACIER study and the comparison of ACI scores between these two groups of older individuals. Furthermore, to extend our understanding of ACI in more ecological contexts, we have already collected acceleration data from older participants performing 10 min free walks in an urban environment, which will provide valuable insights into the applicability of ACI in real-world settings. Subsequently, the final phase will use a prospective approach with a longitudinal design and survival analysis. This involves following older participants over a two-year period to assess the association of ACI and other gait measures with time to first and second falls. 

By addressing these future directions, the ACIER study can significantly contribute to the development of novel and reliable gait assessment tools using readily available wearable sensors, ultimately aiding in fall prevention strategies and improving mobility monitoring in older adults.

## Figures and Tables

**Figure 1 sensors-24-07427-f001:**
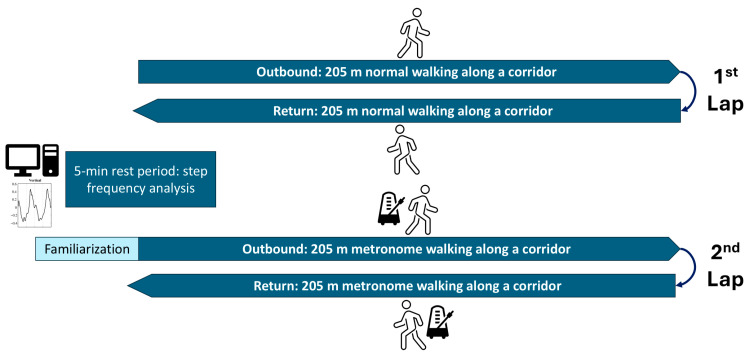
Experimental protocol for normal and metronome walking assessment: two-lap corridor test.

**Figure 2 sensors-24-07427-f002:**
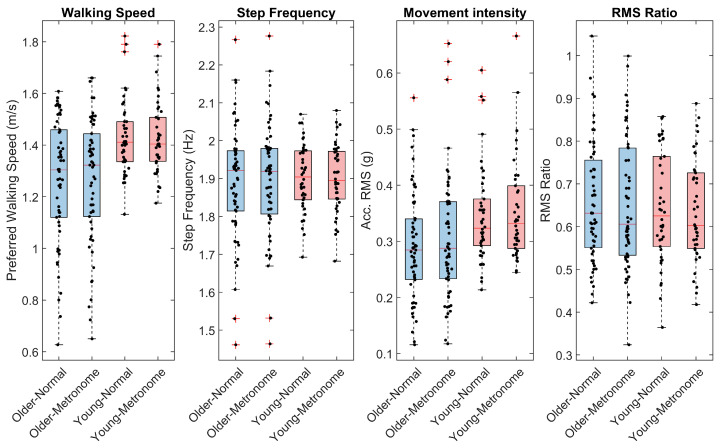
Descriptive statistics of basic gait parameters, movement intensity, and RMS ratio. Sixty older and 42 young adults performed 4 × 200 m indoor walking tests with and without synchronizing their steps to an isochronous metronome at their preferred cadence and walking speed. Box plots show median, quartiles, range of data, and outliers (red crosses) representing values exceeding 1.5 times the interquartile range beyond Q1 and Q3. Individual data are shown as black dots. Average walking speed was measured by displacement timing. Step frequency was assessed by spectral analysis of the acceleration signal. Movement intensity is the RMS of the norm of the 3D acceleration. RMS ratio is the ratio between the mediolateral and the norm of acceleration, which is indicative of the lateral gait stability.

**Figure 3 sensors-24-07427-f003:**
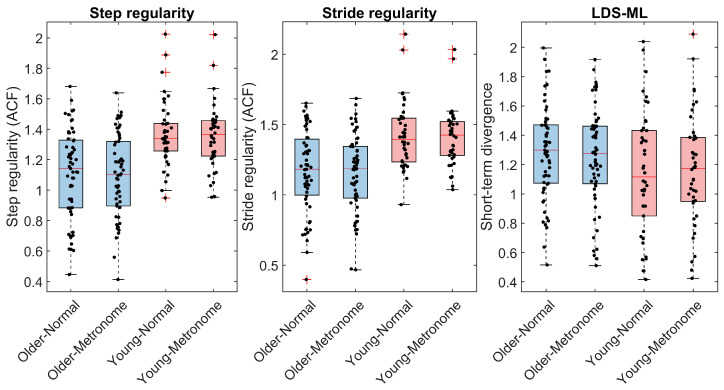
Descriptive statistics of the gait regularity and stability. Sixty older and 42 young adults performed 4 × 200 m indoor walking tests with and without synchronizing their steps to an isochronous metronome at their preferred cadence and walking speed. Box plots show median, quartiles, range of data, and outliers (red crosses) representing values exceeding 1.5 times the interquartile range beyond Q1 and Q3. Individual data are shown as black dots. The autocorrelation function (ACF) method was used to assess the step regularity and the stride regularity. Short-term logarithmic divergence exponents (maximal Lyapunov exponents) of the mediolateral (ML) acceleration, representative of the local dynamic stability (LDS), were assessed using Rosenstein’s algorithm.

**Figure 4 sensors-24-07427-f004:**
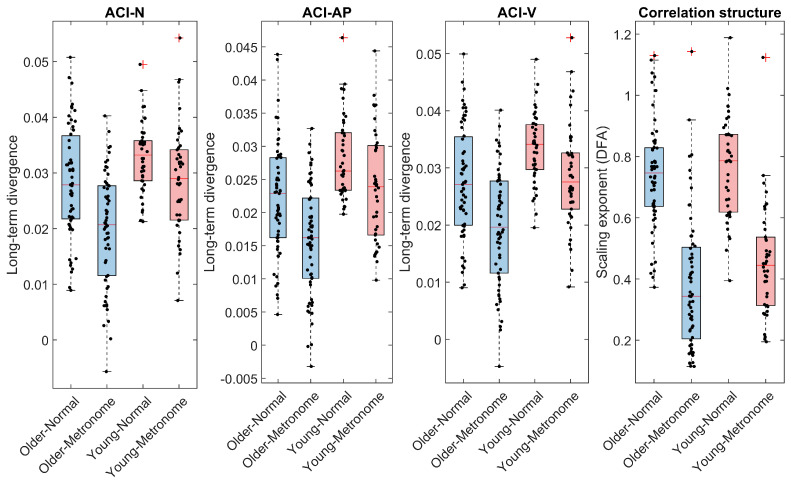
Descriptive statistics of the attractor complexity index (ACI) and the gait complexity (DFA). Sixty older and 42 young adults performed 4 × 200 m indoor walking tests with and without synchronizing their steps to an isochronous metronome at their preferred cadence and walking speed. Box plots show median, quartiles, range of data, and outliers (red crosses) representing values exceeding 1.5 times the interquartile range beyond Q1 and Q3. Individual data are shown as black dots. Long-term logarithmic divergence exponents (maximal Lyapunov exponents) of the vector norm (N), the anteroposterior (AP), and the vertical (V) accelerations, representative of ACI, were assessed using Rosenstein’s algorithm. Scaling exponents (α, correlation structure) were computed based on the stride intervals measured by the foot-mounted accelerometer. The detrended fluctuation analysis (DFA) was applied.

**Figure 5 sensors-24-07427-f005:**
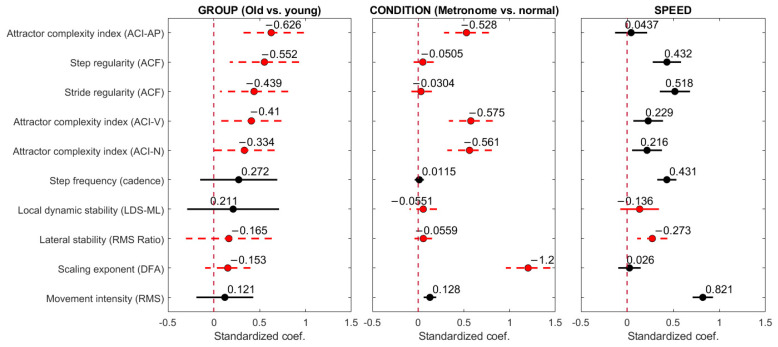
Inferential statistics: mixed-effect linear models. Sixty older and 42 young adults performed 4 × 200 m indoor walking tests with and without synchronizing their steps to an isochronous metronome at their preferred cadence and walking speed. Ten multiple regression models were fitted to the gait metrics obtained from the walking tests with the lower back accelerometer and the foot accelerometer (scaling exponent only). Two independent categorical variables were introduced: group membership (older or young) and walking conditions (normal or metronome walking). In addition, the preferred walking speed was introduced as a continuous covariate. The data were standardized. The absolute values of the regression coefficients (fixed effects) and their 99% confidence intervals are presented graphically, with negative coefficients drawn in red and with dashed lines. The values of the coefficients are added on the top of each line. ACI: attractor complexity index; ACF: autocorrelation function; LDS: local dynamic stability; DFA: detrended fluctuation analysis; RMS: root mean square; N: norm; AP: anteroposterior; V: vertical; ML: mediolateral.

**Table 1 sensors-24-07427-t001:** Summary of the gait metrics used in the study.

Gait Metrics		Principles and Methodology	Applications in Free-Living Conditions
Basic gait parameters	Walking speed	Natural pace measured by timing over the 200 m corridor.	N/A
	Step frequency (SF)	Mean number of steps per second. Computed from the vertical acceleration spectrum via fast Fourier transform (FFT) [59].	[31,32,60]
Variability parameters (lumbar accelerometer)	Movement intensity (RMS)	RMS quantifies the magnitude of a varying signal as the square root of the average of the squared values over a period. Representative of the average amplitude of the acceleration during walking. Calculated using the vector magnitude of the 3D acceleration signals [61].	[31,32,60]
	Lateral stability (RMS ratio)	RMS ratio represents the ratio between RMS in the mediolateral direction and the RMS vector magnitude [62]. It attenuates the dependence of RMS to speed and is thought to be sensitive to impaired dynamic balance [56,62].	[63]
	Step regularity (ACF)	Autocorrelation function (ACF) analyzes cyclic patterns in acceleration signals by comparing values with time-shifted versions, with peak values indicating dominant periods. Higher peaks indicate a pronounced similarity across successive cycles. Step regularity corresponds to the first dominant period. Stride regularity corresponds to the second dominant period [64].	[25,31,32,33]
	Stride regularity (ACF)		
	Local dynamic stability (LDS)	LDS assesses the resilience of gait to perturbations. It is determined by calculating the logarithmic divergence rate between adjacent trajectories within a reconstructed attractor that reflects the gait dynamics (Rosenstein’s algorithm) [65,66,67].	[31,32,60]
	Attractor complexity index (ACI)	ACI has been empirically validated as a surrogate measure for the correlation structure between successive strides. Its calculation follows the same principles as LDS [34,52,53].	[60]
Foot accelerometer	Scaling exponent α (DFA)	Detrended fluctuation analysis (DFA) of stride interval time series provides the scaling exponent (alpha, α), a measure of the correlation structure of gait [68].	N/A

**Table 2 sensors-24-07427-t002:** Descriptive statistics of the normal walking condition.

	Normal Walking									
		Older Participants			Young Participants			Effect Size	Confidence Intervals	
		N	Mean	SD	N	Mean	SD	g	CI Low	CI High
Basic gait parameters	Walking speed (m/s)	58	1.27	0.24	42	1.43	0.15	**−0.80**	**−1.25**	**−0.36**
	Step frequency (Hz)	59	1.89	0.15	42	1.90	0.09	−0.11	−0.60	0.38
Variability measures	Movement intensity (g)	59	0.29	0.10	42	0.35	0.09	**−0.58**	**−1.12**	**−0.08**
	RMS ratio	59	0.66	0.14	42	0.65	0.13	0.08	−0.39	0.6
	Step regularity	59	1.11	0.29	42	1.37	0.22	**−0.97**	**−1.49**	**−0.54**
	Stride regularity	59	1.17	0.30	42	1.42	0.24	**−0.91**	**−1.39**	**−0.47**
Local dynamic stability	LDS-ML	59	1.29	0.32	42	1.15	0.42	0.38	−0.15	0.99
Attractor complexity index	ACI-N	59	0.028	0.010	42	0.033	0.006	**−0.53**	**−1.06**	**−0.07**
	ACI-AP	59	0.022	0.009	42	0.028	0.006	**−0.77**	**−1.33**	**−0.31**
	ACI-V	59	0.027	0.010	42	0.033	0.006	**−0.69**	**−1.23**	**−0.24**
Foot accelerometer	Scaling exponent (DFA)	60	0.74	0.17	42	0.77	0.17	−0.19	−0.73	0.31

Sample size (N), mean, standard deviation (SD), standardized effect size (Hedges’ g) of the difference between age groups, and 99% confidence interval (CI) of the effect size. Results in bold are statistically significant. RMS: root mean square; LDS: local dynamic stability; ACI: attractor complexity index; DFA: detrended fluctuation analysis.

**Table 3 sensors-24-07427-t003:** Descriptive statistics of the metronome walking condition.

	Metronome Walking									
		Older Participants			Young Participants			Effect Size	Confidence Intervals	
		N	Mean	SD	N	Mean	SD	g	CI Low	CI High
Basic gait parameters	Walking speed (m/s)	58	1.26	0.24	42	1.42	0.14	**−0.77**	**−1.23**	**−0.34**
	Step frequency (Hz)	59	1.90	0.15	42	1.90	0.09	−0.06	−0.55	0.46
Variability measures	Movement intensity (g)	58	0.31	0.11	42	0.35	0.09	−0.46	−1.03	0.03
	RMS ratio (%)	58	0.65	0.15	42	0.64	0.12	0.11	−0.41	0.64
	Step regularity (N/A)	58	1.09	0.27	42	1.35	0.21	**−1.02**	**−1.57**	**−0.57**
	Stride regularity (N/A)	58	1.15	0.28	42	1.41	0.20	**−1.01**	**−1.51**	**−0.57**
Local dynamic stability	LDS-ML	58	1.25	0.34	42	1.16	0.38	0.25	−0.26	0.81
Attractor complexity index	ACI-N	58	0.020	0.010	42	0.029	0.010	**−0.87**	**−1.37**	**−0.38**
	ACI-AP	58	0.016	0.008	42	0.024	0.01	**−0.95**	**−1.47**	**−0.46**
	ACI-V	58	0.020	0.010	42	0.028	0.01	**−0.92**	**−1.46**	**−0.42**
Foot accelerometer	Scaling exponent (DFA)	60	0.39	0.22	42	0.46	0.18	−0.33	−0.91	0.18

Sample size (N), mean, standard deviation (SD), standardized effect size (Hedges’ g) of the difference between age groups, and 99% confidence interval (CI) of the effect size. Results in bold are statistically significant. RMS: root mean square; LDS: local dynamic stability; ACI: attractor complexity index; DFA: detrended fluctuation analysis.

**Table 4 sensors-24-07427-t004:** Inferential statistics.

		Multiple Mixed-Effects Regression Models (Fixed Effects)					
		Group (Older vs. Young)			Condition (Normal vs. Metronome)		
		Coef.	CI Low	CI High	Coef.	CI Low	CI High
Basic gait parameters	Walking speed	**−0.167**	**−0.28**	**−0.06**	0.000	−0.015	0.016
	Step frequency	−0.012	−0.078	0.054	0.002	−0.006	0.010
Variability measures	Movement intensity	**−0.056**	**−0.104**	**−0.007**	**0.013**	**0.003**	**0.023**
	RMS ratio	0.013	−0.057	0.083	−0.001	−0.021	0.008
	Step regularity	**−0.253**	**−0.379**	**−0.118**	−0.013	−0.043	0.025
	Stride regularity	**−0.249**	**−0.295**	**−0.109**	−0.009	−0.051	0.010
Local dynamic stability	LDS-ML	0.111	−0.068	0.290	−0.017	−0.076	0.043
Attractor complexity index	ACI-N	**−0.0063**	**−0.0102**	**−0.0024**	**−0.0067**	**−0.0095**	**−0.0038**
	ACI-AP	**−0.0069**	**−0.0102**	**−0.0035**	**−0.0055**	**−0.0079**	**−0.0030**
	ACI-V	**−0.0071**	**−0.0110**	**−0.0033**	**−0.0067**	**−0.0094**	**−0.0039**
Foot accelerometer	Scaling exponent (DFA)	−0.047	−0.113	0.018	**−0.335**	**−0.406**	**−0.264**

Gait metrics were introduced as dependent variables. Group membership (young or older) and condition (normal or metronome) were introduced as categorical independent variables. Regression coefficients (coef.) are presented with their 99% confidence intervals. Results in bold are statistically significant. RMS: root mean square; LDS: local dynamic stability; ACI: attractor complexity index; DFA: detrended fluctuation analysis.

## Data Availability

The datasets generated and analyzed during the current study are openly available in the Zenodo repository at https://doi.org/10.5281/zenodo.10148824. These data can be freely accessed and used for replication, further analysis, or comparative studies in accordance with the terms of the repository.

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
