# Peer review of "Validity of Linear and Nonlinear Measures of Gait Variability to Characterize Aging Gait with a Single Lower Back Accelerometer"

_sensors, 2024, doi:10.3390/s24237427_

Round 1
Reviewer 1 Report
Comments and Suggestions for Authors
The study is accurately reported and the results are well presented and discussed. I personally think that it can be published in the Journal after slight and minor revisions.
- It has been reported that the effects of a concurrent task on gait might differ in younger subjects [1] than in older people [2]. I think that this should be discussed more, together with an explanation on why the only cognitive distortion that has been inserted is the metronome walking.
- Lines 65-74. Machine learning has been used to simplify the extraction of parameters from IMU in clinical settings [3-4]. These new approaches could solve the problems reported in this paragraph.
- A graphical representation of the study design and protocol can be useful.
- Some details of the Methods are repeated throughout the section
- Why not using also metrics of smoothness [5-6]
[1] Caramia, Carlotta, et al. "Smartphone-based answering to school subject questions alters gait in young digital natives." Frontiers in public health 8 (2020): 187.
[2] Yogev-Seligmann G, Hausdorff JM, Giladi N. The role of executive function and attention in gait. Mov Disord. (2008) 23:329–42. doi: 10.1002/mds.21720
[3] Bibbo, Daniele, et al. "Machine learning to detect, stage and classify diseases and their symptoms based on inertial sensor data: A mapping review." Physiological Measurement (2023).
[4] Sharifi Renani, Mohsen, et al. "Deep learning in gait parameter prediction for OA and TKA patients wearing IMU sensors." Sensors 20.19 (2020): 5553.
[5] Balasubramanian S, Melendez-Calderon A, Roby-Brami A, Burdet E. On the analysis of movement smoothness. J Neuroeng Rehabil. (2015) 12:112. doi: 10.1186/s12984-015-0090-9
[6] Beck Y, Herman T, Brozgol M, Giladi N, Mirelman A, Hausdorff JM. SPARC: a new approach to quantifying gait smoothness in patients with Parkinson's disease. J Neuroeng Rehabil. (2018) 15:49. doi: 10.1186/s12984-018-0398-3
Reviewer 2 Report
Comments and Suggestions for Authors
The study investigates the use of the Attractor Complexity Index (ACI) as a tool for evaluating gait variability, comparing it against traditional gait metrics under normal and metronome-paced walking conditions. The authors hypothesize that ACI, derived from nonlinear dynamics, can capture attentional demands and identify aging-related gait changes with a single accelerometer.
The introduction provides an overview of gait variability and aging, it could benefit from more detailed references and context. Specifically, adding more background on motor-cognitive interference and attentional demands in aging could enhance the understanding of ACI's significance.
The methods are mostly clear, but a few clarifications are needed. For example, consider briefly explaining ACI's intra-session reliability to support its robustness as a metric. Additionally, a short description of any preprocessing steps for the accelerometer data would be helpful (specifically sections 2.4 and 2.5).
The results are well-organized, but certain figures could be enhanced for readability. For example, figures showing comparisons between age groups under different walking conditions could be annotated more clearly to highlight key findings (Figures 1–7).
The study presents ACI as a metric sensitive to attentional demands. However, the findings on ACI’s responsiveness to metronome walking may need a more nuanced interpretation. It would be helpful to discuss possible alternative explanations, such as synchronization effects rather than attentional demand alone, to provide a balanced view of ACI’s capabilities (Discussion section, particularly in the discussion on ACI and metronome walking).
Suggesting additional experiments or applications on ACI would strengthen its impact. For example, a brief mention of how ACI could be tested in real-world, unsupervised settings or with varied cognitive tasks would provide useful insights into its practical applications.
The references are generally appropriate, but a few additional citations could improve the theoretical foundation, particularly regarding cognitive-motor interference theories and attentional demands in gait analysis.
Comments on the Quality of English LanguageThe English is generally clear, but refining certain sections would enhance readability. For instance, simplifying technical phrases in the Introduction (Page 1–2) could make the paper more accessible to a wider audience.
A careful review of sentence structure in these sections would improve readability. For example, breaking down longer sentences into shorter, more direct statements would enhance flow and clarity. This would especially help in the sections describing statistical methods (Page 9) and results interpretation (Page 14).
Terms like "gait complexity" and "gait regularity" are used interchangeably in some places, which may confuse readers. Ensuring consistent terminology, particularly in discussing gait variability metrics (Pages 6–7), would provide clearer insights into ACI and its relationship with other measures.
